# CXXC5 mediates growth plate senescence and is a target for enhancement of longitudinal bone growth

Sehee Choi[1,2], Hyun-Yi Kim[1,2], Pu-Hyeon Cha[1,2], Seol Hwa Seo[1,2], Chulho Lee[1,2], Yejoo Choi[1,2], Wookjin Shin[1,2], Yunseok Heo[1,3], Gyoonhee Han[1,2], Weontae Lee[1,3], Kang-Yell Choi[1,2,4]

Longitudinal bone growth ceases with growth plate senescence during puberty. However, the molecular mechanisms of this phenomenon are largely unexplored. Here, we examined Wnt-responsive genes before and after growth plate senescence and found that CXXC finger protein 5 (CXXC5), a negative regulator of the Wnt/$\beta$-catenin pathway, was gradually elevated with reduction of Wnt/$\beta$-catenin signaling during senescent changes of rodent growth plate. $Cxxc5^{-/-}$ mice demonstrated delayed growth plate senescence and tibial elongation. As CXXC5 functions by interacting with dishevelled (DVL), we sought to identify small molecules capable of disrupting this interaction. In vitro screening assay monitoring CXXC5–DVL interaction revealed that several indirubin analogs were effective antagonists of this interaction. A functionally improved indirubin derivative, KY19382, elongated tibial length through delayed senescence and further activation of the growth plate in adolescent mice. Collectively, our findings reveal an important role for CXXC5 as a suppressor of longitudinal bone growth involving growth plate activity.

## Introduction

Longitudinal bone growth takes place in the growth plate, which is composed of a thin layer of transient cartilage tissue. Chondrocytes in this cartilage layer proliferate and undergo hypertrophic differentiation followed by apoptosis and subsequent remodeling into bone tissue, resulting in bone elongation (Kronenberg, 2003). Longitudinal bone growth occurs rapidly during fetal development and early childhood, but then slows and, eventually ceases at the end of puberty with growth plate senescence (Nilsson & Baron, 2004; Lui et al, 2011). Presently, many children undergo early pubertal development with growth plate senescence occurring sooner. These phenomena, known as precocious puberty, reveal premature termination of longitudinal bone growth, resulting in short adult stature

(Carel et al, 2004). However, the underlying mechanisms that regulate growth plate senescence are largely unknown.

In recent years, accumulating evidence from basic and clinical studies revealed that chondrocyte activity and status is directly subject to regulation by paracrine signaling within the growth plate (Baron et al, 2015). Specifically, Wnt/$\beta$-catenin signaling has emerged as a key player in growth plate maturation, and mutation of genes involved in the regulation of Wnt/$\beta$-catenin signaling often resulted in impaired bone growth. For example, cartilage-specific loss of $Ctnnb1$ encoding $\beta$-catenin caused defects in longitudinal bone growth (Chen et al, 2008; Dao et al, 2012). In addition, treatment with an inhibitor of glycogen synthase kinase $3\beta$ (GSK3$\beta$), a serine/threonine kinase that destabilizes $\beta$-catenin (Doble & Woodgett, 2003), resulted in tibial elongation in the ex vivo culture system (Gillespie et al, 2011). Furthermore, a large meta-analysis of genome-wide association studies identified 423 loci that contribute to common variation in adult human height and found genes involved in the Wnt/$\beta$-catenin pathway such as $AXIN2$, $WNT4$, and $CTNNB1$ (Wood et al, 2014).

CXXC finger protein 5 (CXXC5) is a negative regulator of Wnt/$\beta$-catenin signaling, functioning via interaction with the PDZ domain of dishevelled (DVL) in the cytosol (Andersson et al, 2009; Kim et al, 2010, 2015). Inhibition of the CXXC5–DVL interaction improved several pathophysiological phenotypes involving Wnt/$\beta$-catenin signaling, including osteoporosis, cutaneous wounds, and hair loss through activation of the Wnt/$\beta$-catenin signaling (Kim et al, 2015, 2016; Lee et al, 2015, 2017).

In this study, we found that CXXC5 expression progressively increased in the resting, proliferative, and hypertrophic chondrocytes undergoing growth plate senescence. We also found that estrogen, a sex hormone that is elevated during the pubertal period, induced CXXC5 expression followed by decrement of $\beta$-catenin in chondrocytes. Furthermore, $Cxxc5^{-/-}$ mice displayed enhanced chondrocyte proliferation and differentiation in the late pubertal growth plate as well as longer tibiae at adulthood. These results suggest that CXXC5 contributes to growth plate senescence at puberty. Small molecules that activate the Wnt/$\beta$-catenin pathway

[1]Translational Research Center for Protein Function Control, Yonsei University, Seoul, Korea   [2]Department of Biotechnology, College of Life Science and Biotechnology, Yonsei University, Seoul, Korea   [3]Department of Biochemistry, College of Life Science and Biotechnology, Yonsei University, Seoul, Korea   [4]CK Biotechnology Inc, Seoul, Korea

Correspondence: kychoi@yonsei.ac.kr

by inhibiting the CXXC5–DVL interaction were obtained by the use of an in vitro screening system monitoring fluorescent intensity that reveals binding of the PTD-DBMP (protein transduction domain fused DVL binding motif peptide), which contains sequence of CXXC5 binding to DVL and is conjugated to fluorescein isothiocyanate (FITC), onto PZD domain of DVL (Kim et al, 2016). Interestingly, several GSK3$\beta$ inhibitors, including 6-bromoindirubin-3'-oxime (BIO) and indirubin-3'-oxime (I3O), were identified as initial hits. A functionally improved indirubin derivative, KY19382, was obtained by chemical synthesis and effectively inhibited both GSK3$\beta$ kinase activity and CXXC5–DVL interaction. These functions were confirmed by kinetic measurement of GSK3$\beta$ enzyme activity and in vitro CXXC5–DVL binding, respectively. Therefore, KY19382 effectively activated Wnt/$\beta$-catenin signaling via dual functions: initial activation by inhibition of GSK3$\beta$ and subsequently further enhancement of the signaling by interference of CXXC5–DVL interaction. We further demonstrated that KY19382 markedly enhanced proliferation and differentiation of chondrocytes and induced longitudinal tibiae growth in adolescent mice by delaying growth plate senescence.

In summary, CXXC5, a protein induced with pubertal progression in the growth plate chondrocytes, is characterized as a key factor mediating termination of longitudinal bone growth by promoting growth plate senescence. A small molecule targeting both CXXC5–DVL interaction and GSK3$\beta$ activity delayed growth plate senescence through the effective activation of Wnt/$\beta$-catenin signaling. Altogether, our results support a small molecular approach targeting CXXC5 as a potential therapeutic strategy for treatment of children with growth retardation attributed to early growth plate senescence.

# Results

### CXXC5 expression progressively increases in the growth plate at later stages of puberty

To elucidate the involvement of Wnt/$\beta$-catenin signaling in growth plate senescence, we used gene set enrichment analysis and investigated the expression profiles of Wnt-responsive genes in the proliferative zone of 3-wk-old (pre- and early puberty) and 12-wk-old (early adulthood) rats (Gene Expression Omnibus [GEO]: GSE16981). We found that the signatures of Wnt/$\beta$-catenin signaling-activated genes were significantly down-regulated in the growth plates of the 12-wk-old rats (Fig 1A). Moreover, the mRNA level of Cxxc5, a negative regulator of Wnt/$\beta$-catenin signaling, was gradually elevated during pubertal progression (GEO: GSE16981) (Fig 1B), showing a statistically significant increase at 12 wk compared to other inhibitors of Wnt/$\beta$-catenin signaling (Apcdd1, Cxxc4, Dkk2, Igfbp4, Sfrp family, Shisa family, Sost, and Wif1) (Fig S1). However, CXXC4, a structural and functional analog of CXXC5 that also functions as a negative regulator of Wnt/$\beta$-catenin signaling (Hino et al, 2001; Katoh & Katoh, 2004), was not significantly induced at puberty in the growth plate zones of humans or rats when compared with CXXC5 (Fig S2A and B). To examine the pubertal period in more detail, the growth plates of proximal tibiae from 3-, 6-, 9-, and 12-wk-old mice were collected and subjected to additional analyses. Up-regulation of Cxxc5 were confirmed by quantitative real-time PCR (qRT-PCR) analyses at the later

stages of pubertal progression compared to 3-wk-old mice (Fig 1C). Immunoblot analyses also showed that CXXC5 gradually increased with the decrement of $\beta$-catenin and chondrogenic markers including COL2A1, RUNX2, COL10A1, and MMP13 in the growth plates of mice undergoing pubertal progression (Fig 1D). The inverse correlation between CXXC5 and Wnt/$\beta$-catenin signaling was verified by immunohistochemical (IHC) analyses showing progressive increase of cytosolic CXXC5 with the gradual decrease of nuclear $\beta$-catenin in all growth plate zones of 3- to 12-wk-old mice (Fig 1E). Next, we confirmed the inhibitory effects of CXXC5 on Wnt/$\beta$-catenin pathway and overall chondrogenic maturation at the cell level. Along with reduction of the WNT3A-inducd Wnt/$\beta$-catenin signaling target genes (Axin2 and Wisp1) by Cxxc5 overexpression, we observed that WNT3A stimulated transcription of the signaling molecules required for chondrocyte maturation (Fgf18, Pthlh encoding PTHrP, and Ihh). Moreover, diverse chondrogenic markers (Col2a1, Sox9, Runx2, Alp, and Mmp13) were suppressed by Cxxc5 overexpression (Fig S3).

### CXXC5 mediates growth plate senescence induced by the sexual hormone, estrogen

Induction of CXXC5 during pubertal progression suggests its involvement in growth plate senescence at puberty. As estrogen, a hormone involved in sexual maturation, is elevated at puberty and known to play a role in growth plate senescence (Weise et al, 2001), we examined the effect of 17$\beta$-estradiol (E$_2$), a major estrogenic hormone in the circulation, on CXXC5 expression in the human chondrocyte cell line, C28/I2. Treatment of E$_2$ induced expression of CXXC5 in a time-dependent manner, achieving a maximal level at 24 h $\beta$-Catenin level was reduced after 24 h of E$_2$ treatment (Fig 2A). As shown by immunocytochemical analysis, E$_2$ prominently elevated cytosolic CXXC5 and repressed cytosolic and nuclear $\beta$-catenin (Fig 2B). The role of E$_2$ on growth plate senescence was further confirmed by the use of an ex vivo tibial culture system that demonstrated reduced tibial length with decreased height of proliferative and hypertrophic zones in the growth plate after E$_2$ treatment (Fig 2C and D). The induction of cytosolic CXXC5 and the decrement of nuclear $\beta$-catenin in the chondrocytes of E$_2$-treated growth plates supports the previously identified relationship between growth plate senescence and inactivation of Wnt/$\beta$-catenin signaling (Fig 2E versus Fig 1E). To verify the involvement of estrogen in CXXC5 expression and growth plate senescence, the effects of E$_2$ treatment were compared in 6-wk-old Cxxc5$^{+/+}$ and Cxxc5$^{-/-}$ mice. E$_2$-induced structural senescence of the tibial growth plate was shown in Cxxc5$^{+/+}$ mice with increment of CXXC5 expression in the whole growth plate zones but was hardly observed in Cxxc5$^{-/-}$ mice (Fig 2F). In addition, there were no significant changes in BrdU incorporation and $\beta$-catenin expression in E2-treated Cxxc5$^{-/-}$ mice (Fig 2F). These results show that CXXC5 mediates growth plate senescence upon induction by estrogen.

### CXXC5 plays a key role in suppression of longitudinal bone growth at late puberty

To further define the role of CXXC5 in growth plate senescence during pubertal progression, we assessed longitudinal bone growth and growth plate senescence in Cxxc5$^{+/+}$ and Cxxc5$^{-/-}$ mice. Cxxc5$^{-/-}$

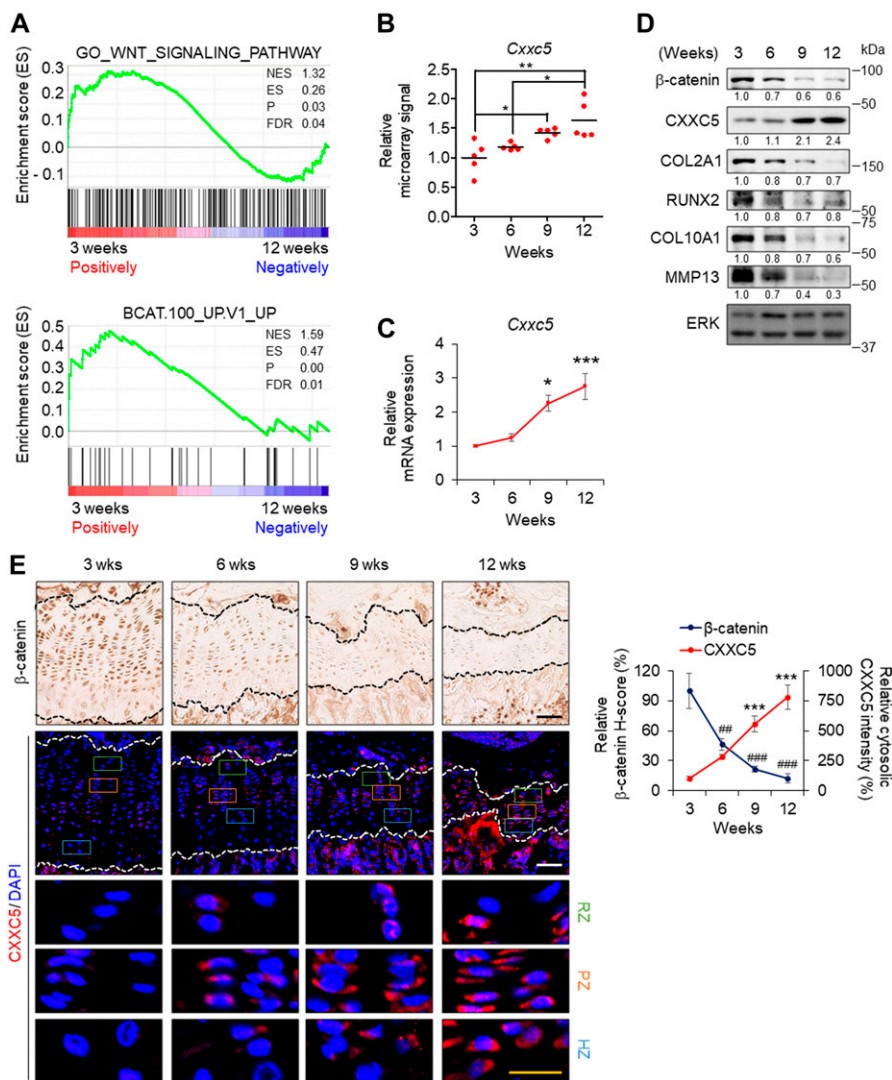

**Figure 1.  Changes in Wnt/β-catenin pathway and CXXC5 expression during growth plate senescence.**
**(A)** Gene set enrichment analysis (GSEA) of microarray transcriptome data from the proliferative zone of growth plates in 3- and 12-wk-old rats (GEO: GSE16981) for Wnt/β-catenin signaling–activated gene signatures (upper, MSigDB: M11722 and lower, MSigDB: M2680) (n = 5). **(B)** The relative expression changes of *Cxxc5* in the proliferative zone 3-, 6-, 9-, and 12-wk-old rat growth plates (GEO: GSE16981) (mean ± SEM, n = 5, ANOVA, *P* = 1.88 × 10$^{-3}$; Tukey's post-hoc test, **P* < 0.05 and ***P* < 0.005). **(C)** qRT-PCR analyses of relative mRNA expression of *Cxxc5* in the growth plate of proximal tibiae of 3-, 6-, 9-, and 12-wk-old mice (mean ± SEM, n = 5, ANOVA, *P* = 1.57 × 10$^{-4}$; Tukey's post-hoc test, **P* < 0.05 and ****P* < 0.0005 versus 3-wk-old). **(D)** Immunoblot analyses with the indicated antibodies were performed in the growth plate of proximal tibiae of 3-, 6-, 9-, and 12-wk-old mice. **(E)** IHC analyses with the indicated antibodies in the growth plate of proximal tibiae of 3-, 6-, 9-, and 12-wk-old mice (left) and quantitative analyses (mean ± SEM, n = 6, ANOVA, *P* = 8.09 × 10$^{-7}$ for CXXC5 expression, *P* = 1.19 × 10$^{-5}$ for β-catenin expression; Tukey's post-hoc test, ##*P* < 0.005, *** or ###*P* < 0.0005 versus 3-wk-old) (right). Black and white scale bars, 50 μm. Yellow scale bar, 20 μm. NES, normalized enrichment score; ES, enrichment score; FDR, false discovery rate; RZ, resting zone; PZ, proliferative zone; HZ, hypertrophic zone.

mice showed significantly enhanced tibial lengths at 12 wk of age (Fig 3A and B). With aging, growth plates of *Cxxc5*$^{+/+}$ mice naturally underwent structural senescence as monitored by gradual reduction of the height of resting, proliferative, and hypertrophic zones with a concomitant decline in the number of chondrocytes in each zone (Fig 3C–E). However, these age-related changes were significantly delayed in *Cxxc5*$^{-/-}$ mice, although the growth plates of *Cxxc5*$^{-/-}$ mice did eventually undergo structural senescence with aging (Fig 3C–E). The retardation of growth plate senescence by *Cxxc5* deletion was further supported by marked increases of Ki67 and β-catenin protein levels together with *Runx2* mRNA level in chondrocytes of the growth plates of 11-wk-old *Cxxc5*$^{-/-}$ mice compared with 11-wk-old *Cxxc5*$^{+/+}$ mice (Fig 3F). The activation of Wnt/β-catenin signaling and the promotion of chondrogenic differentiation was further confirmed by the up-regulation of Wnt/β-catenin target genes (*Axin2*, *Fosl1*, and *Wisp1*) and chondrogenic markers (*Col2a1*, *Col10a1*, *Alp*, and *Runx2*) in the growth plates of 9-wk-old *Cxxc5*$^{-/-}$ mice compared with 9-wk-old *Cxxc5*$^{+/+}$ mice (Fig 3G).

As CXXC5 functions as a negative regulator of Wnt/β-catenin pathway by binding to DVL (Andersson et al, 2009), we tested whether a PTD-DBMP, which interferes with the CXXC5–DVL interaction (Kim et al, 2015), would exert effects similar to the loss of *Cxxc5* on growth plate senescence. Indeed, injection of the PTD-DBMP into the growth plates of 7-wk-old mice (late puberty) (Fig 3H) increased the number of resting, proliferative, and hypertrophic chondrocytes per column with induction of β-catenin and RUNX2 levels in chondrocytes of the growth plate (Fig 3I and J). Overall, these results indicate that CXXC5 plays a role in the structural senescence of the growth plate, which can be acquired by inhibition of the CXXC5–DVL interaction.

### KY19382 activates Wnt/β-catenin signaling through inhibitory effects on both CXXC5–DVL interaction and GSK3β activity

To identify small molecules that mimic the function of the PTD-DBMP and delay growth plate senescence, we screened 2,280 compounds from chemical libraries (1,000 from ChemDiv and

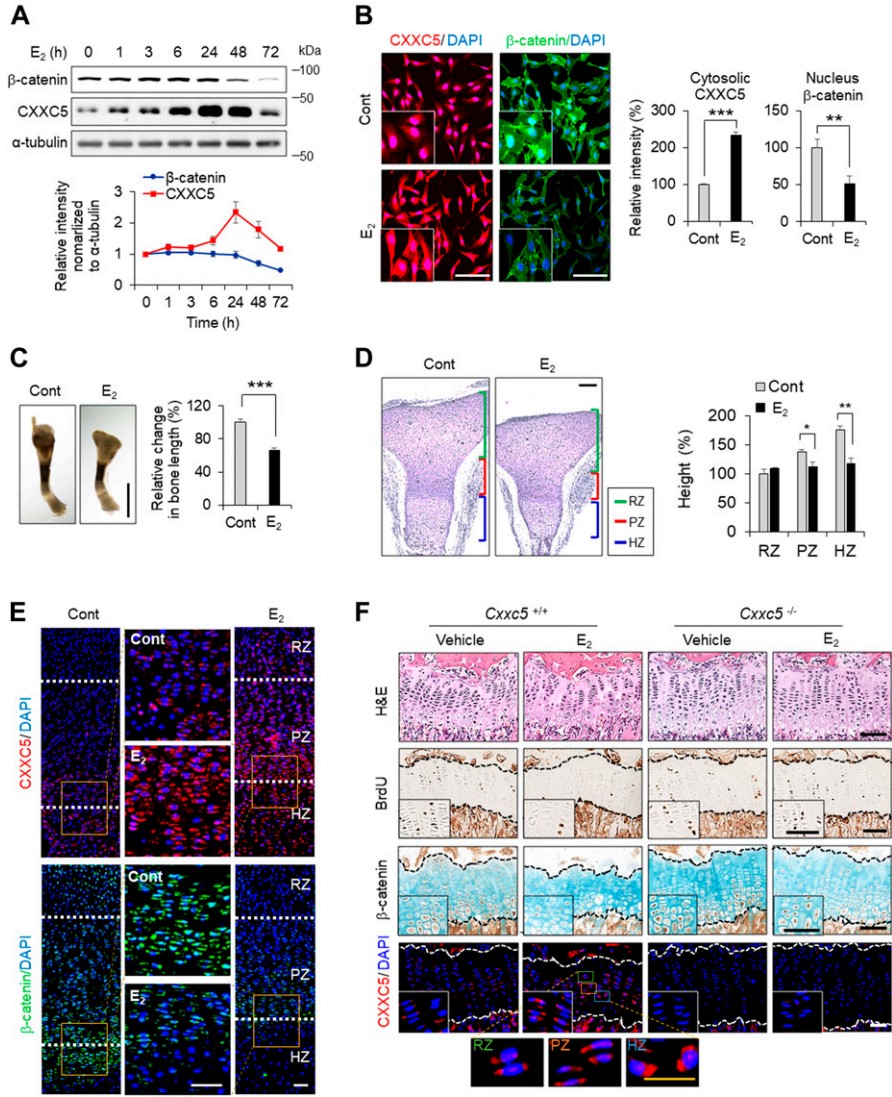

**Figure 2. The effects of estrogen on CXXC5 expression and the mediation of growth plate senescence.**
**(A)** Immunoblotting (upper) and quantitative analyses (mean ± SEM, n = 3) (lower) in C28/I2 cells treated with 100 nM $E_2$ (17$\beta$-estradiol) for 0, 1, 3, 6, 24, 48, or 72 h. **(B)** Immunocytochemical staining (left) and quantitative analyses of the fluorescent intensity (mean ± SEM, n = 3; $t$ test, **$P < 0.005$ and ***$P < 0.0005$) (right) in C28/I2 cells treated with 100 nM $E_2$ for 40 h. Scale bars, 100 $\mu$m. **(C–E)** Tibial organ cultures (E15.5) incubated with 100 nM $E_2$ for 6 d. Representative images at 6 d (C, left) and quantitative analyses of the growth changes (mean ± SEM, n = 5; $t$ test, ***$P < 0.0005$) (C, right). Scale bar, 1 mm. H&E staining (D, left) and quantification of each zone height (mean ± SEM, n = 5; $t$ test, *$P < 0.05$ and **$P < 0.005$) (D, right) in the growth plate. Scale bar, 200 $\mu$m. IHC analyses of $\beta$-catenin and CXXC5 **(E)**. Scale bars, 50 $\mu$m. **(F)** 3-wk-old $Cxxc5^{+/+}$ and $Cxxc5^{-/-}$ mice were treated with $E_2$ cypionate (70 $\mu$g/kg) by i.m. injection once a week for 3 wk (n = 3–4). Representative images of H&E staining and IHC analyses for BrdU, $\beta$-catenin, and CXXC5 in the growth plates of proximal tibiae are shown. The area within the dashed lines indicates the growth plate zone. Black and white scale bars, 50 $\mu$m. Yellow scale bar, 20 $\mu$m. RZ, resting zone; PZ, proliferative zone; HZ, hypertrophic zone.

1,280 from SigmaLOPAC) with an in vitro assay system that monitors the CXXC5–DVL interaction (Kim et al, 2016) (Fig S4). In this screening system, we identified the indirubin analogs BIO (compound 8) and I3O (compound 12), which are known GSK3$\beta$ inhibitors (Meijer et al, 2003), as top-ranked positive initial hits (Tables S1 and S2). As the indirubin derivatives contain indole rings, which are known to interact with the PZD domain of DVL (Mahindroo et al, 2008), BIO and I3O also interacted with DVL PDZ domain (Protein Data Bank [PDB]: 2KAW) in an in silico docking modeling (Fig S5A and B).

To obtain functionally improved compound, 60 indirubin derivatives were newly synthesized by replacing the functional groups at the $R_1$ and $R_2$ sites of the indirubin backbone based on the structure of BIO and I3O (Fig S6A). By evaluating them for in vitro CXXC5–DVL binding activity, in vitro GSK3$\beta$ kinase activity, and TOPFlash Wnt reporter activity, we obtained 5, 6-dichloroindirubin-3'-methoxime (KY19382; Fig 4A) as an optimal compound for further investigation; KY19382 markedly inhibited both in vitro CXXC5–DVL

interaction ($IC_{50}$ of KY19382 = $1.9 \times 10^{-8}$ M; Fig 4B) and in vitro GSK3$\beta$ activity ($IC_{50}$ of KY19382 = $1 \times 10^{-8}$ M; Fig 4C) with the strong enhancement of the TOPFlash Wnt reporter activity (Fig 4D).

We further characterized the possible binding sites for KY19382 on the DVL PDZ domain (PDB: 2KAW) using the in silico docking program (Fig S6B). Structural simulations of the KY19382–DVL PDZ complex revealed that residues involved in the interaction with KY19382 were similar to the DBMP-binding sites (Kim et al, 2016). Compared with BIO or I3O, the estimated binding energy for the KY19382–DVL PDZ complex was improved (BIO = $-81.80$ kcal·mol$^{-1}$ or I3O = $-75.34$ kcal·mol$^{-1}$ versus KY19382 = $-97.96$ kcal·mol$^{-1}$) (Fig S5A and B versus Fig S6B).

The role of KY19382 in the activation of Wnt/$\beta$-catenin signaling was further verified by the increment of $\beta$-catenin with the inactivation of GSK3$\alpha$/$\beta$ (Fig 4E) and the interruption of the CXXC5–DVL interaction (Fig 4F), resulting in the elevated nuclear translocation of $\beta$-catenin in ATDC5 cells (Fig 4G). This efficient induction of $\beta$-catenin by KY19382 treatment is likely dependent on both the

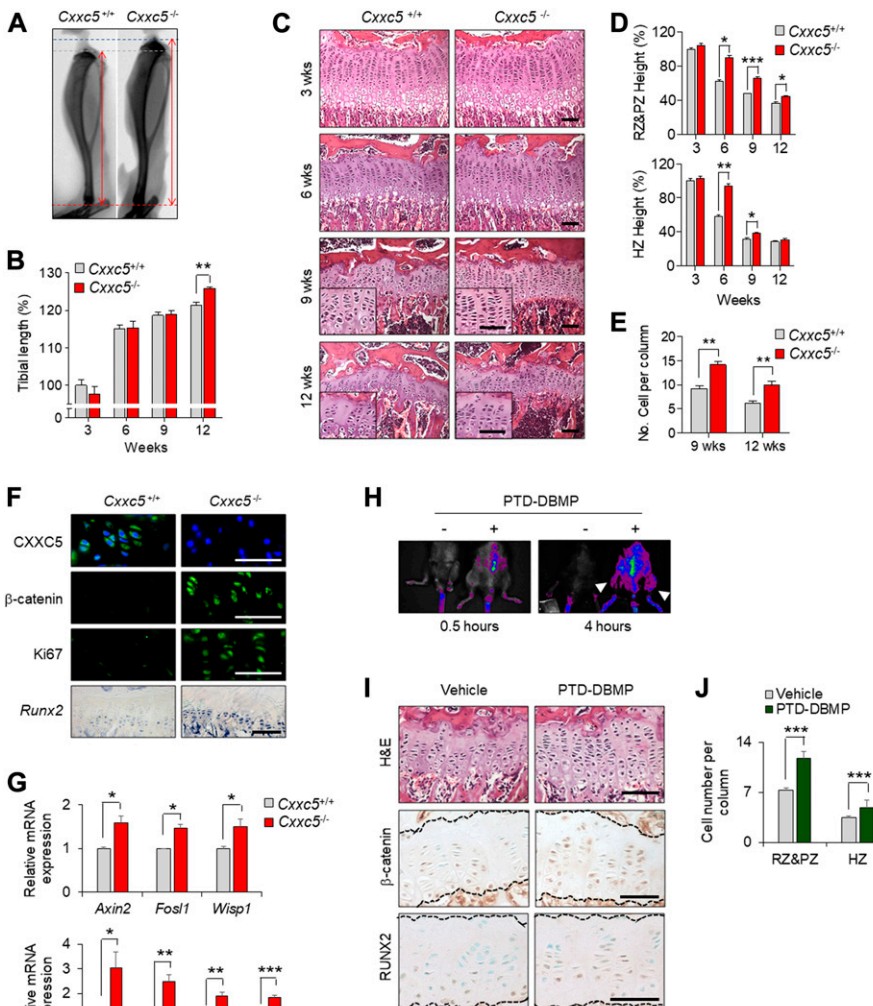

**Figure 3. Functional characterization of CXXC5 in growth plate senescence as an inhibitor of the Wnt/β-catenin pathway via interaction with DVL.**
**(A)** Representative radiographs of tibiae of 12-wk-old *Cxxc5*⁺/⁺ and *Cxxc5*⁻/⁻ mice. **(B)** Tibial length of 3-, 6-, 9-, and 12-wk-old *Cxxc5*⁺/⁺ and *Cxxc5*⁻/⁻ mice were measured (mean ± SEM, n = 4–10 mice per group; ANOVA, $P = 2.37 \times 10^{-2}$; Bonferroni's post-hoc test, **$P < 0.005$). **(C–E)** H&E staining (C) and quantitative analyses of each zone height (D) in the growth plate of proximal tibiae of 3-, 6-, 9-, and 12-wk-old *Cxxc5*⁺/⁺ and *Cxxc5*⁻/⁻ mice (mean ± SEM, n = 3–5 mice per group; ANOVA, $P = 2.9 \times 10^{-7}$ for upper panel, $P = 1.92 \times 10^{-9}$ for lower panel; Bonferroni's post-hoc test, *$P < 0.05$, **$P < 0.005$, and ***$P < 0.0005$). Quantitative analyses of the cell number per column in the growth plates of 9- and 12-wk-old *Cxxc5*⁺/⁺ and *Cxxc5*⁻/⁻ mice (mean ± SEM, n = 5, **$P < 0.005$) (E). **(F)** IHC analyses with the indicated antibodies or in situ hybridization for *Runx2* in the proximal tibial growth plates of 11-wk-old *Cxxc5*⁺/⁺ and *Cxxc5*⁻/⁻ mice. **(G)** qRT-PCR analyses of mRNA levels of Wnt-target genes and chondrogenic markers in the growth plate of proximal tibiae of 9-wk-old *Cxxc5*⁺/⁺ and *Cxxc5*⁻/⁻ mice (mean ± SEM, n = 3; t test, *$P < 0.05$, **$P < 0.005$, and ***$P < 0.0005$). **(H–J)** The PTD-DBMP (1 mg/kg) were administered to 7-wk-old mice by daily i.p. injection for 2 wk (n = 10). In vivo fluorescent imaging shows the presence of the PTD-DBMP in the treated mice **(H)**. White arrowheads indicate the growth plate regions of tibia. H&E staining, IHC analyses for β-catenin and RUNX2 in the growth plates of proximal tibiae were performed (I). Quantitative analyses of the cell number in the RZ, PZ, and HZ of growth plates (mean ± SEM, n = 3; t test, *$P < 0.05$ and **$P < 0.005$) (J). Scale bars, 50 μm.

inactivation of GSK3α/β and the interruption of the CXXC5–DVL interaction.

### KY19382 delays growth plate senescence and promotes longitudinal bone growth

To investigate the effects KY19382 on growth plate senescence, 0.1 mg/kg KY19382 was intraperitoneally injected into the growth plates of 7-wk-old mice (late puberty) daily for 2 wk. The total growth plate height, monitored by COL2A1 immunostaining, was significantly increased by KY19382 treatment (Fig 5A). This effect was confirmed by increased numbers of both proliferative and hypertrophic chondrocytes per column, assessed by BrdU- and RUNX2-positive cells, respectively (Fig 5A–C). Along with these effects, nuclear β-catenin was dramatically increased in the growth plate chondrocytes by KY19382 treatment (Fig 5A). Immunoblot analyses also showed that KY19382 increased β-catenin and chondrogenic markers, such as COL2A1, RUNX2, and MMP13, in the growth plate (Fig 5E). These functional and structural changes

demonstrate the ability of KY19382 in delaying growth plate senescence.

Next, we tested the effects of KY19382 in rapidly growing young mice by administering 0.1 mg/kg KY19382 at 3 wk of age (early puberty) daily for 2 wk. With the increase of total growth plate height, as evidenced by COL2A1 expression, the height of each growth plate zone and BrdU-positive cells were elevated in KY19382-treated mice (Fig 5F–H). As observed in older mice, β-catenin–expressing chondrocytes were also increased by KY19382 treatment (Fig 5F). To exclude the possibility that the expanded HZ was a result of delayed cartilage resorption (Chen et al, 2015), we performed TRAP staining in tibiae sections. The number of TRAP-positive foci in the growth plate/trabecular interface was not different between the groups (Fig 5I), indicating that KY19382 did not affect the cartilage resorption of rapidly growing young mice. However, older mice treated with KY19382 from 7 wk of age to 9 wk of age exhibited elevated TRAP-positive foci compared to vehicle-treated mice (Fig 5A and D). These effects showed that the overall process of growth plate maturation, including preparation of the space to be replaced

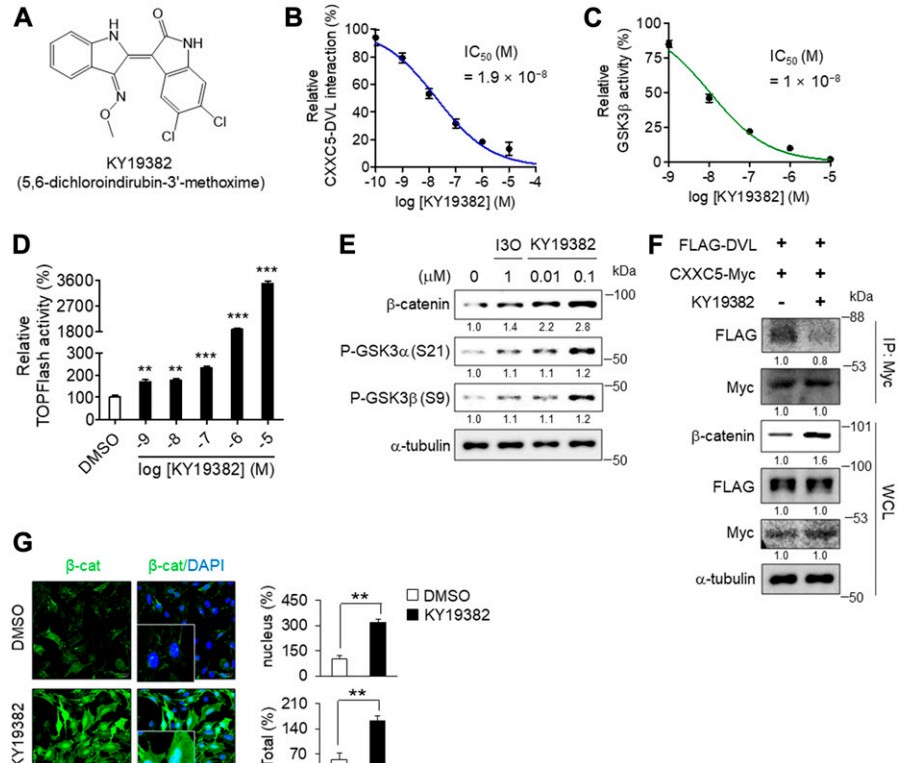

**Figure 4. Identification of functional properties of KY19382 in activating the Wnt/β-catenin pathway.**
**(A)** Chemical structure of KY19382. **(B)** In vitro binding assay to analyze the effect of KY19382 on CXXC5–DVL interaction (mean ± SEM, n = 3). The $IC_{50}$ value was determined from the dose–response curve. **(C)** In vitro kinase assay to analyze the effect of KY19382 on kinase activity of GSK3β (mean ± SEM, n = 3). The $IC_{50}$ value was determined from the dose–response curve. **(D)** Analyses of TOPFlash activity in HEK293 reporter cells grown with the indicated concentrations of KY19382 for 18 h (mean ± SEM, n = 4; t test, **P < 0.005 and ***P < 0.0005 versus DMSO-treated control). **(E)** Immunoblot analyses with the indicated antibodies in ATDC5 cells treated with I3O or KY19382 for 24 h. **(F)** Immunoblot analyses of whole cell lysates immunoprecipitated with anti-Myc in ATDC5 cells treated with 0.1 μM KY19382 for 4 h after transfection with pCMV-FLAG-DVL1 and pcDNA3.1-CXXC5-Myc. **(G)** Immunocytochemical staining (left) and quantitative analyses (mean ± SEM, n = 3; t test, **P < 0.005) (right) for β-catenin in ATDC5 cells treated with 0.1 μM KY19382 for 48 h. Scale bar, 100 μm.

by osteoblastic bone formation (Usmani et al, 2012), was activated by KY19382 treatment in spite of the senescent growth plate of late pubertal mice.

The role of KY19382 on chondrocyte proliferation was further verified in vitro by the enhanced number of BrdU–positive ATDC5 cells after KY19382 treatment (Fig S7A). In addition, the mRNA levels of chondrogenic markers were up-regulated by KY19382 in ATDC5 and C28/I2 cells (Fig S7B and C). Importantly, these effects were abolished by siRNA–mediated *Ctnnb1* knockdown (Fig S7B). We also explored off-target effects of KY19382 by measuring mRNA levels of target genes for various signaling pathways in KY19382-treated ATDC5 cells (Fig S8). Although KY19382 markedly increased the expression levels of Wnt/β-catenin target genes, such as *Wisp1* and *Axin2*, the 19 other genes that respond to other pathways were not significantly altered. These results demonstrate that KY19382 promotes chondrocyte proliferation and differentiation via specific activation of the Wnt/β-catenin pathway.

To investigate the comprehensive effects from pre- and early puberty to the adulthood period, we performed long-term administration of KY19382 for 10 wk in mice from the age of 3 to 13 wk. Daily treatment of 0.1 mg/kg KY19382 significantly increased the length of tibiae compared with the vehicle-treated group (Fig 5J). In addition, no histological abnormalities were detected in the articular cartilage and the liver tissues of KY19382-treated mice (Fig S9A and B). During the 10 wk of treatment, no difference in weight was observed among the groups (Fig S9C). Taken together, these data reveal that KY19382 induces longitudinal bone growth by promoting growth plate maturation in rapidly growing young mice

as well as delaying growth plate senescence in older mice, without noticeable toxicity. Furthermore, the pharmacokinetic evaluation of KY19382 displayed a relatively favorable intraperitoneal bio-availability (F = 16.74%), showing half-life of 16.20 h and an exposure level of 6,555.79 ng·hr/ml (Table S3).

## Discussion

The hallmark of growth plate senescence includes a decline in the overall height of the growth plate with a decrease in the number of resting, proliferative, and hypertrophic chondrocytes per column and an increase in the spacing between adjacent chondrocyte columns. Unlike "senescence," which generally refers to specific cellular program, the term "growth plate senescence" indicates a physiological loss of function that occurs with increasing age (Gafni et al, 2001; Nilsson & Baron, 2004). Although many children undergo early growth plate senescence and reach a short height in adult-hood because of precocious puberty, the mechanism of these phenomena is poorly understood. Recent studies suggest that growth plate activity is primarily regulated by paracrine factors that directly exert their function on chondrocytes within the growth plate (Lui et al, 2011; Baron et al, 2015). Wnt/β-catenin signaling has been implicated in these functions; however, the molecular mechanisms and factors controlling growth plate senescence are still unexplored.

In this study, we identified that a negative feedback regulator of the Wnt/β-catenin pathway, CXXC5, gradually increased with

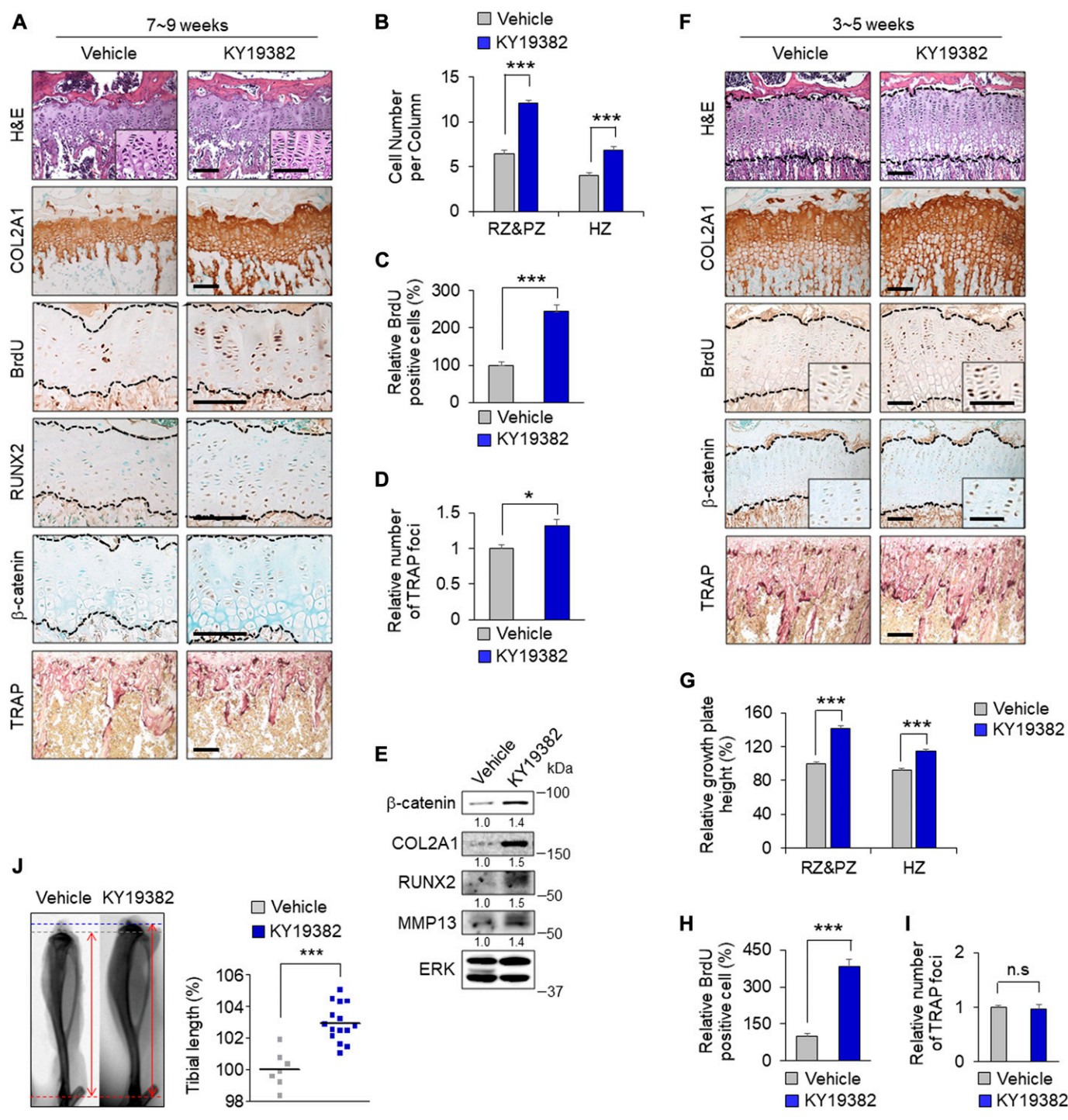

**Figure 5. The effects of KY19382 on growth plate senescence and longitudinal bone growth.**
**(A–I)** KY19382 (0.1 mg/kg) was administered to 7-wk-old mice (A–E) or 3-wk-old mice (F–I) by daily intraperitoneal injection for 2 wk (n = 7). H&E staining, IHC analyses with the indicated antibodies, and TRAP staining in the growth plates of proximal tibiae treated with KY19382 (A, F). Quantitative analyses of the cell number per column (mean ± SEM, n = 7; *t* test, ***P < 0.0005) **(B)** or the height (mean ± SEM, n = 7; *t* test, ***P < 0.0005) (G) of resting zone and proliferative zone (RZ&PZ) and hypertrophic zone (HZ) in the growth plates of proximal tibiae. Quantitative analyses of BrdU-positive cells in the growth plates (mean ± SEM, n = 5; *t* test, ***P < 0.0005) (C, H). Quantitative analyses of the number of TRAP-positive foci along 250 µm of the cartilage/bone interface (mean ± SEM, n = 3; *t* test, *P < 0.05) (D, I). Immunoblot analyses with the indicated antibodies in the growth plate of proximal tibiae of mice treated with KY19382 (E). Scale bars, 50 µm. **(J)** 3-wk-old mice were intraperitoneally injected with KY19382 (0.1 mg/kg) daily for 10 wk. Representative radiographs are shown (left), and tibial length was measured (right) (mean ± SEM, n = 7–15; *t* test, ***P < 0.0005). The area within the dashed lines indicates the growth plate zone. n.s., no significance; TRAP, tartrate-resistant acid phosphatase.

suppression of β-catenin in all growth plate zones at the later stages of puberty. Moreover, we found that CXXC5 is induced by estrogen, a sex hormone that increases with pubertal progression. This finding is supported by previous work demonstrating that CXXC5 is a direct target of estrogen signaling in a breast cancer cell line (Yasar et al, 2016). Although estrogen is known to trigger senescent changes of the growth plate and its deficiency in both male and female humans results in non-fused growth plate and continual bone elongation (Bilezikian et al, 1998; Juul, 2001; Vandenput & Ohlsson, 2009; Rochira et al, 2015), the mediators and signaling pathways exerting these effects of estrogen on the growth plate senescence have not been illustrated. Here, we observed the abolishment of estrogen-derived growth plate senescence in $Cxxc5^{-/-}$ mice and further characterized a role of CXXC5 as a mediator in the estrogen-induced growth plate senescence and subsequent termination of longitudinal bone growth. The function of CXXC5 is exerted to inhibit of Wnt/β-catenin signaling, as shown by in vivo studies that correlates with the inverse relationship of the expression patterns of CXXC5 and β-catenin in the chondrocytes.

Our observation that CXXC5 expression is increased in all growth plate zones with repression of Wnt/β-catenin signaling during pubertal progression indicates that CXXC5 plays an important role in suppression of overall chondrogenic processes, including chondrocyte proliferation and hypertrophic differentiation. Furthermore, the role of CXXC5 in the regulation of overall chondrogenesis is supported by suppression of the signaling molecules involved in growth plate maturation, such as FGF18, IHH, and PTHrP (Kronenberg, 2003; Long & Ornitz, 2013), with the inactivation of Wnt/β-catenin signaling by CXXC5.

CXXC5 can localize to the cytosol or the nucleus, depending on cell type and tissue (Kim et al, 2014; Lee et al, 2015). The specific induction of cytosolic CXXC5 during growth plate senescence supports that CXXC5 function is acquired by binding with DVL in the cytosol. Unlike *CXXC5*, *CXXC4*, a protein structurally and functionally similar to CXXC5 (Hino et al, 2001; Katoh & Katoh, 2004), was not significantly expressed in the growth plate during pubertal progression, indicating that CXXC5 plays a specific role in growth plate senescence.

As cytosolic CXXC5 functions via interaction with PDZ domain of DVL, we validated the CXXC5–DVL interaction as a target for the development of drugs that delay growth plate senescence with the use of the PTD-DBMP, a CXXC5–DVL blocking peptide. To further develop small molecules capable of inducing longitudinal bone growth by delaying growth plate senescence, we screened small molecular libraries using an in vitro screening system that monitors the CXXC5–DVL interaction (Kim et al, 2016). The indirubin analogs, BIO and I3O, which are known GSKβ inhibitors that activate Wnt/β-catenin signaling (Meijer et al, 2003), were screened as potential CXXC5–DVL inhibitors. Development of functionally improved indirubin derivatives, especially KY19382, confirmed that these family compounds can have the dual roles as inhibitors of both CXXC5–DVL interaction and GSK3β activity. Our results showed that KY19382 effectively increased the longitudinal growth of tibiae by delaying growth plate senescence through the accompanying promotion of chondrocyte proliferation and differentiation. The high effectiveness of KY19382 in enhancing longitudinal bone growth may be due to dual functions via both enhancement of

growth plate maturation in the rapidly growing young period by inactivation of GSK3β and delay of growth plate senescence in the late pubertal period by interference of CXXC5–DVL interaction.

KY19382 did not reveal significant off-target effects as observed by the lack of significant activation of 19 other pathway-specific genes, except the Wnt/β-catenin pathway-target genes. Furthermore, we did not observe any adverse effects on articular cartilage after administration of 0.1 mg/kg KY19382, which induced longitudinal bone growth. Our strategy of targeting cytosolic CXXC5, which functions via the interaction with DVL, also prevents the undesirable functioning of nuclear CXXC5 as a transcription factor.

Overall, estrogen-induced CXXC5 during pubertal progression is a critical factor that promotes growth plate senescence and inhibits longitudinal bone growth, by exerting its functions via inactivation of Wnt/β-catenin signaling (Fig 6A). An effective small molecular approach that activates Wnt/β-catenin signaling via a dual mechanism of inhibition of GSK3β and disruption of CXXC5–DVL interaction is a novel therapeutic strategy for children with growth retardation that involves early growth plate senescence (Fig 6B).

# Materials and Methods

## Cell culture and reagents

The mouse chondrogenic cell line, ATDC5, was obtained from the RIKEN Cell Bank. The human juvenile costal chondrocyte cell line, C28/I2, was provided by Dr. W U Kim (Catholic University, Korea). HEK293-TOP cells (HEK293 cells containing the chromosomally incorporated TOPFlash gene) were provided by Dr. S Oh (Kuk Min University, Korea). ATDC5 cells were maintained in DMEM/F12 (1:1) (Gibco) supplemented with 5% FBS (Gibco). To induce hypertrophic differentiation, ATDC5 cells were incubated with insulin–transferrin–sodium selenite supplement (Gibco) in three-dimensional alginate beads for 3 d, as described previously (Kawasaki et al, 2008). C28/I2 and HEK293-TOP cells were maintained in DMEM (Gibco) containing 10% FBS. All chemicals were dissolved in dimethyl sulfoxide (DMSO; Sigma-Aldrich) for the in vitro studies. For $E_2$ (17β-estradiol; Sigma-Aldrich) treatment, the cells were cultured in phenol red–free DMEM/F12 with 5% charcoal-stripped FBS for 24 h followed by serum-free medium for 24 h before the experiment. The PTD-DBMP was synthesized by Peptide 2.0 Inc.

## Plasmids, siRNAs, and transfection

The plasmids pcDNA3.1-CXXC5-Myc and GFP-CXXC5 have been previously described (Kim et al, 2015). The pCMV-FLAG-DVL1 was provided by Dr. EH Jho (Seoulsirip University, Korea).

The following siRNA sequences were used for ATDC5 cells: *Ctnnb1* (encoding β-catenin) siRNA-1, sense AUUACAAUCCGGUUGUGAA-CGUCCC and anti-sense GGGACGUUCACAACCGGAUUGUAAU; *Ctnnb1* siRNA-2, sense UAAUGAAGGCGAACGGCAUUCUGGG and anti-sense CCCAGAAUGCCGUUCGCCUUCAUUA.

Lipofectamine (Invitrogen) was used for plasmid transfection and RNAiMax (Invitrogen) was used for siRNA transfection, according to the manufacturer's instructions.

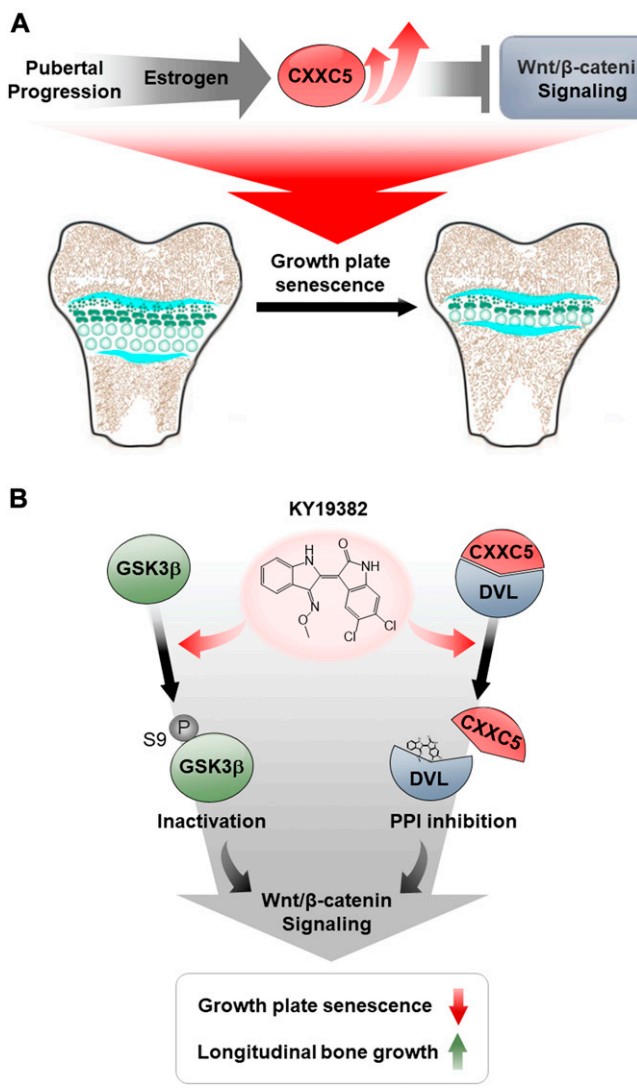

**Figure 6. Schematic representation of the role of CXXC5 and molecular mechanism of KY19382 in longitudinal bone growth.**
**(A)** A proposed model for the role of CXXC5 in the growth plate. With pubertal progression, estrogen, which increases during sexual maturation, induces CXXC5 expression and subsequently inhibits the Wnt/$\beta$-catenin pathway, resulting in growth plate senescence. **(B)** A working model of KY19382 for the stimulation of longitudinal bone growth. In activating Wnt/$\beta$-catenin signaling, KY19382 functions as a dual-targeting compound by 1) inactivating GSK3$\beta$ and 2) inhibiting CXXC5–DVL interaction, which results in the delaying of growth plate senescence and the promotion of longitudinal bone growth. PPI, protein–protein interaction.

## Animals

*Cxxc5*$^{-/-}$ mice were established in a previous study (Kim et al, 2015). To manipulate growth plate senescence by estrogen, 3-wk-old *Cxxc5*$^{+/+}$ and *Cxxc5*$^{-/-}$ male mice received weekly i.m. injections of either 70 $\mu$g/kg estradiol (E$_2$) cypionate (Sigma-Aldrich) or vehicle (cottonseed oil) for 3 wk. To investigate the effects of KY19382 treatment on longitudinal bone growth, C57BL/6 male mice were purchased from KOATECH (Gyeonggido, Korea). KY19382 (0.1 mg/kg) was administered daily by i.p. injection to 3- and 7-wk-old mice for 2 wk or to 3-wk-old mice for 10 wk. For BrdU labeling experiments, the

mice were i.p. injected with 50 mg/kg BrdU (Sigma-Aldrich) before 24 h to euthanize. All animal procedures were approved by the Institutional Animal Care and Use Committee of Yonsei University (Korea) and conducted based on the guidelines of the Korean Food and Drug Administration.

### Radiographic and histochemical analyses

Plain radiographs were taken using an X-ray apparatus (KODAK DXS 4000 Pro SYSTEM; Carestream Health). The tissues were fixed in 4% PFA, decalcified in 10% EDTA (pH 7.4), dehydrated, embedded in paraffin, and sectioned to 4-$\mu$m thickness (Leica Microsystems). The tissues sections were rehydrated and used for further analyses, including H&E, TRAP, and IHC staining. To perform IHC staining, the sections were incubated with citrate buffer (pH 6.0) at 80°C for 30 min, with 0.05% trypsin working solution (pH 7.8) for 30 min at 37°C, or with 0.5% pepsin (Sigma-Aldrich) for 15 min at 37°C. Then, the sections were blocked with 5% normal goat serum (Vector Laboratories) and 0.3% Triton X-100 in PBS for 1 h at room temperature. For 3,3'-diaminobensidine (DAB) staining, the sections were incubated with 0.345% H$_2$O$_2$ for 15 min. Before incubating the sections with mouse primary antibody, mouse IgG was blocked using a M.O.M kit (Vector Laboratories). The sections were incubated at 4°C overnight with the following primary antibodies: anti–$\beta$-catenin (#610154; 1:50; BD Bioscience), anti-CXXC5 (1:200; lab-made) anti-BrdU (M0744; 1:200; DAKO), anti-COL2A1 (PA5-11462; 1:100; Thermo Fisher Scientific), anti-Ki67 (ab15580; 1:200; Abcam), and anti-RUNX2 (ab23981; 1:200; Abcam). Then, the sections were incubated at room temperature for 1 h with biotinylated anti-mouse (BA-9200; 1:200; Vector Laboratories) or biotinylated anti-rabbit (BA-1000; 1:200; Vector Laboratories) secondary antibodies. The sections were then incubated in avidin–biotin complex solutions (Vector Laboratories), stained with a DAB kit (Vector Laboratories) for 3–30 min, and counterstained with methyl green (Sigma-Aldrich). All incubations were conducted in humid chambers. Staining was observed with an ECLIPSE TE2000-U microscope (Nikon). For fluorescence staining, the sections were incubated with primary antibody at 4°C overnight, followed by incubation with anti-mouse Alexa Fluor 488 (A11008; 1:200; Thermo Fisher Scientific) or anti-rabbit Alex Fluor 555 (A21428; 1:200; Thermo Fisher Scientific) secondary antibodies at room temperature for 1 h. The sections were then counterstained with DAPI (Sigma-Aldrich) for 5 min and mounted in Gel/Mount media (BioMeda Corporation). All incubations were conducted in dark humid chambers. The fluorescent signals were visualized using an LSM700 META confocal microscope (Carl Zeiss Inc) at excitation wavelengths of 488 nm (Alexa Fluor 488), 543 nm (Alexa Fluor 555), and 405 nm (DAPI).

### Immunocytochemistry

ATDC5 or C28/I2 cells were seeded on glass coverslip in 12-well culture plates. The cells were washed with PBS and fixed with 4% PFA at room temperature for 15 min. After permeabilization with 0.1% Triton X-100 for 15 min and blocking with 5% BSA for 1 h, the cells were incubated with primary antibodies specific for $\beta$-catenin (1:100) or CXXC5 (1:200) at 4°C overnight. The cells were washed in PBS and incubated with Alexa Fluor 488 or Alexa Fluor 555

secondary antibodies (1:200) at room temperature for 1 h. Cell nuclei were counterstained with DAPI for 10 min and the stained samples were examined under an LSM700 META microscope using 405-, 488-, or 543-nm excitation wavelengths. For BrdU assay, the cultured cells were incubated with BrdU solution (25 $\mu$M) overnight, followed by immunocytochemical staining with antibody against BrdU (1:100).

### Immunoblot analyses

The cells were washed with ice-cold PBS and tissues were ground with a mortar and pestle in liquid nitrogen before lysis in RIPA buffer (150 mM NaCl, 50 mM Tris, pH 7.4, 1% NP-40, 0.25% sodium deoxycholate, 1 mM EDTA, protease inhibitors, and phosphatase inhibitors). Protein samples were separated on an 8–12% SDS–PAGE and transferred to a nitrocellulose membrane (Whatman). Immunoblotting was performed with the following primary antibodies: anti–$\beta$-catenin (sc-7199; 1:3,000; Santa Cruz Biotechnology), anti-CXXC5 (lab made; 1:200) anti-Myc tag (M192-3; 1:1,000; MBL), anti-FLAG (F7425; 1:1,000; Sigma-Aldrich), anti-p-GSK3$\alpha/\beta$ (S21/S9; Cell Signaling Technology, 9331; 1:1,000), anti-COL2A1 (sc-28887; 1:500; Santa Cruz Biotechnology), anti-RUNX2 (ab23981; 1:500; Abcam), anti-COL10A1 (LSL-LB-0092; 1:500; Cosmo Bio), anti-MMP13 (sc-30073; 1:500; Santa Cruz Biotechnology), anti-ERK (sc-94; 1:3,000; Santa Cruz Biotechnology), and anti–$\alpha$-tubulin (3873S; 1:20,000; Cell Signaling Technology). The samples were then incubated with horseradish peroxidase–conjugated anti-mouse (7076; 1:3,000; Cell Signaling Technology), anti-rabbit (1706515; 1:3,000; Bio-Rad), or anti-goat (sc-2020; 1:3,000; Santa Cruz Biotechnology) secondary antibodies. Protein bands were visualized with ECL (Amersham Bioscience) using a luminescent image analyzer, LAS-3000 (Fujifilm). Immunoblot bands were analyzed using Multi-Gauge V3.0 software (Fujifilm). Points of interest from immunoblot bands were marked and quantified using densitometry, and the background signals were subtracted from respective immunoblot signals. Relative densitometry values were presented as the intensity ratios of each protein to loading control protein ($\alpha$-tubulin or ERK).

### Immunoprecipitation

Immunoprecipitation was performed as previously described (Kim et al, 2015). To monitor the protein–protein interactions, 1 mg of WCLs were incubated with anti-DVL1 and protein G agarose beads (GenDEPOT) or anti-Myc and protein A agarose beads (GenDEPOT) at 4°C for 16 h, and the beads were then washed three times in RIPA buffer. The resulting immune complexes were resolved by SDS–PAGE, and immunoblotting was performed with the indicated antibodies.

### Tibial organ culture

Tibiae were isolated from embryonic day 15.5 (e15.5) mice and cultured for 6 d in phenol red–free $\alpha$-MEM (Gibco) containing ascorbic acid, $\beta$-glycerophosphate, BSA, L-glutamine, and penicillin–streptomycin, as previously described (Gillespie et al, 2011). After dissection, tibiae were incubated in medium overnight and then treated with E$_2$ (Sigma-Aldrich). Media and reagents were changed every 48 h. Tibial images were captured using an SMZ-745T microscope (Nikon). Tibial length was measured before treatment and after 6 d in culture. The samples were then prepared for paraffin embedding, sectioned, and analyzed by H&E and IHC staining.

### Reporter assay

HEK293-TOP cells were seeded into each well of a 24-well plate. The cells were treated with individual compounds at indicated concentration and cultured for 18 h. The cells were then harvested and lysed in 60 $\mu$l of Reporter Lysis Buffer (Promega) according to the manufacturer's instructions. After centrifugation, 20 $\mu$l of the supernatant was used to measure luciferase activity. Relative luciferase activities were normalized to that of the DMSO-treated control.

### Reverse transcription and qRT-PCR

Total RNA was extracted using Trizol reagent (Invitrogen) according to the manufacturer's instructions. 2 $\mu$g of RNA was reverse-transcribed using 200 units of reverse transcriptase (Invitrogen) in a 40-$\mu$l reaction carried out at 37°C for 1 h. For qRT-PCR analyses, 5–100-fold diluted cDNA (1 $\mu$l) was amplified in 10 $\mu$l reaction mixture containing iQ SYBR Green Supermix (QIAGEN) and 10 pmol of the primer set (Bioneer). The comparative cycle-threshold method was used, and *ACTB*-encoding $\beta$-actin or *GAPDH* served as an endogenous control. The following primer sets were used:

**List of primers used.**

| Gene | Strand | Primer sequences |
|---|---|---|
| Human | | |
| ACTB | F | 5'-AGAGCTACGAGCTGCCTGAC-3' |
| | R | 5'-AGCACTGTGTTGGCGTACA-3' |
| COL2A1 | F | 5'-TGGAAAGCCTGGTGATGATGGTG-3' |
| | R | 5'-TGACCTTTGACACCAGGAAGGC-3' |
| MMP13 | F | 5'-GAAGACCTCCAGTTTGCAGAGC-3' |
| | R | 5'-TTCAGGATTCCCGCGAGATTTG-3' |
| RUNX2 | F | 5'-CACCTTGACCATAACCGTCTTCAC-3' |
| | R | 5'-CATCAAGCTTCTGTCTGTGCCTTC-3' |
| VEGFA | F | 5'-AGGGCAGAATCATCACGAAGTGG-3' |
| | R | 5'-GTCTCGATTGGATGGCAGTAGC-3' |
| Mouse | | |
| Actb | F | 5'-GGATGCAGAAGGAGATTACT-3' |
| | R | 5'-CCGATCCCACACAGAGTACTT-3' |
| Alp | F | 5'-GGGACTGGTACTCGGATAAC-3' |
| | R | 5'-CTGATATGCGATGTCCTTGC-3' |
| Col2a1 | F | 5'-GCCTGTCTGCTTCTTGTAA-3' |
| | R | 5'-TGCGGTTGGAAAGTGTTT-3' |
| Col10a1 | F | 5'-TCCACTCGTCCTTCTCAG-3' |

| Gene | Strand | Primer sequences |
|---|---|---|
| | R | 5'-TTTAGCCTACCTCCAAATGC-3' |
| Ctnnb1 | F | 5'-ACAAGCCACAAGATTACAAGAA-3' |
| | R | 5'-GCACCAATATCAAGTCCAAGA-3' |
| Fgf18 | F | 5'-TGGGGAAGCCTGATGGTACT-3' |
| | R | 5'-CCCTTGGGGTAACGCTTCAT-3' |
| Gapdh | F | 5'-ACCCAGAAGACTGTGGATGG-3' |
| | R | 5'-GGATGCAGGGATGATGTTCT-3' |
| Ihh | F | 5'-CTCTTGCCTACAAGCAGTTCA-3' |
| | R | 5'-CCGTGTTCTCCTCGTCCTT-3' |
| Mmp9 | F | 5'-TGAAGTCTCAGAAGGTGGAT-3' |
| | R | 5'-ATGGCAGAAATAGGCTTTGT-3' |
| Mmp13 | F | 5'-TAAGACACAGCAAGCCAGA-3' |
| | R | 5'-CACATCAGTAAGCACCAAGT-3' |
| Pthlh | F | 5'-GAGATCCACACAGCCGAAAT-3' |
| | R | 5'-CGTCTCCACCTTGTTGGTTT-3' |
| Runx2 | F | 5'-AAGGACAGAGTCAGATTACAGA-3' |
| | R | 5'-GTGGTGGAGTGGATGGAT-3' |
| Sox9 | F | 5'-AACTGGAAACCTGTCTCTCT-3' |
| | R | 5'-ACAACACACGCACACATC-3' |
| Vegfa | F | 5'-TTATTTATTGGTGCTACTGTTTATCC-3' |
| | R | 5'-TCTGTATTTCTTTGTTGCTGTTT-3' |

The primer sets of pathway-specific target genes in Fig S8 were described in a previous study (Kim et al, 2016).

### GSK3β kinase assay

GSK3β (human) was incubated with 8 mM MOPS (pH 7.0), 0.2 mM EDTA, 20 $\mu$M YRRAAVPPSPSLSRHSSPHQS(p) EDEEE (phospho-GS2 peptide), 10 mM Mg acetate, and [γ-33P-ATP] (specific activity ~500 cpm/pmol, concentration as required). The reaction was initiated by the addition of the Mg-ATP mixture. After incubation for 40 min at room temperature, the reaction was stopped by addition of 3% phosphoric acid solution. 10 $\mu$l of the reaction was then spotted onto a P30 filtermat and washed three times for 5 min in 50 mM phosphoric acid and once in methanol before drying and scintillation counting.

### Database

The gene expression profile results were deposited in NCBI's GEO database (http://www.ncbi.nlm.nih.gov/geo/) and are accessible through GEO accession number GSE16981, GSE14007, and GSE9160.

### Quantitation of signal intensity

For DAB immunostaining, validation of the immnohistochemical scoring (H-score) was performed using the automated digital image analysis software ImageJ (National Institutes of Health, Bethesda, MD) and the IHC Profiler plug-in (Varghese et al, 2014). For immunofluorescent staining, the intensity was analyzed with NIS

Elements V3.2 software (Nikon). The blue channel was used as a reference to visualize the nuclei, and the threshold was defined for red, green, or blue channels. Mean intensity was calculated in the red and green channels separately, and mean values were estimated from analyses of three independent experiments.

### Statistical analyses

All data are expressed as the mean ± SEM, and the number of samples is indicated in each figure legend. If not specified in the figure legend, the number of samples was n ≥ 3 for the in vivo, ex vivo, and in vitro experiments, including Western blots, radiographs, and immunohistochemistry. The representative images were those in good agreement with the consistent observation. The data were statistically analyzed by unpaired two-tailed t test for two groups. For more than two groups, we used an ANOVA followed by Tukey's or Bonferroni's post-hoc test. P value < 0.05 were considered to be statistically significant (the Materials and Methods section of the Supplementary Information).

# Supplementary Information

# Acknowledgements

We thank W-U Kim and E-H Jho for providing cells and reagents. This work was supported by the National Research Foundation of Korea grant funded by the Korean Government (MSIP) (grants 2015R1A2A1A05001873, 2016R1A5A1004694, 2019R1A2C3002751). S Choi, H-Y Kim, P-H Cha, SH Seo, and W Shin were supported by a BK21 PLUS program.

## Author Contributions

S Choi: conceptualization, data curation, formal analysis, investigation, visualization, methodology, and writing-original draft, review, and editing.
HY Kim: conceptualization, data curation, formal analysis, and methodology
PH Cha: formal analysis and methodology.
SH Seo: data curation.
C Lee: data curation.
Y Choi: data curation.
W Shin: data curation.
Y Heo: formal analysis.
G Han: formal analysis and supervision.
W Lee: formal analysis and supervision.
KY Choi: formal analysis, supervision, funding acquisition, project administration, and writing-original draft, review, and editing.

## Conflict of Interest Statement

KY Choi is an inventor on patent applications for newly synthesized indirubin derivatives including KY19382 which have been licensed to CK Biotechnology

Inc., a company headquartered in Seoul, Korea. These indirubin derivatives are developing drugs to address unmet clinical needs in diverse pathophysiology. The authors have no further conflicts of interest to declare.

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
