## [Reviewer comments · Life Science Alliance]

Life Science Alliance

CXXC5 mediates growth plate senescence and is a target for enhancement of longitudinal bone growth

Sehee Choi, Hyun-Yi Kim, Pu-Hyeon Cha, Seol Hwa Seo, Chulho Lee, Yejoo Choi, Wookjin Shin, Yunseok Heo, Gyoonhee Han, Weontae Lee, and Kang-Yell Choi

DOI: <https://doi.org/10.26508/lsa.201800254> Corresponding author: Kang-Yell Choi, Yonsei University

Review Timeline:	Submission Date:	2018-11-23
	Editorial Decision:	2018-12-21
	Revision Received:	2019-03-06
	Editorial Decision:	2019-03-25
	Revision Received:	2019-04-01
	Accepted:	2019-04-01

Scientific Editor: Andrea Leibfried

Transaction Report:

December 21, 2018

Re: Life Science Alliance manuscript #LSA-2018-00254-T

Dear Dr. Choi,

Thank you for submitting your manuscript entitled "CXXC5 mediates growth plate senescence and is a target for enhancement of longitudinal bone growth" to Life Science Alliance. The manuscript was assessed by expert reviewers, whose comments are appended to this letter.

As you will see, the reviewers think that your work will be of interest to others once revised. However, they also think that your conclusions are currently not sufficiently supported. Given the reviewers' input, we would like to invite you to submit a revised version of your manuscript. Importantly, while some issues can get addressed by text changes / clarifications, a more defined analysis of the growth plate needs to be performed including better marker usage as well as *cxxc5* expression analysis in the GP (rev1 point 1, 2, 4, 6; rev3 point 2, and also rev2). Furthermore, the robustness of the results should be demonstrated (rev3, point 1) and the effects of the CXXC5 inhibitor should get better analyzed (reviewer #3, points 3 and 4).

Thank you for this interesting contribution to Life Science Alliance. We are looking forward to receiving your revised manuscript.

Sincerely,

Andrea Leibfried, PhD
Executive Editor
Life Science Alliance

Meyerhofstr. 1
69117 Heidelberg, Germany
t +49 6221 8891 502
e a.leibfried@life-science-alliance.org
www.life-science-alliance.org

- A letter addressing the reviewers' comments point by point.
- An editable version of the final text (.DOC or .DOCX) is needed for copyediting (no PDFs).
- High-resolution figure, supplementary figure and video files uploaded as individual files: See our detailed guidelines for preparing your production-ready images, <http://life-science-alliance.org/authorguide>
- Summary blurb (enter in submission system): A short text summarizing in a single sentence the study (max. 200 characters including spaces). This text is used in conjunction with the titles of papers, hence should be informative and complementary to the title and running title. It should describe the context and significance of the findings for a general readership; it should be written in the present tense and refer to the work in the third person. Author names should not be mentioned.

B. MANUSCRIPT ORGANIZATION AND FORMATTING:

Full guidelines are available on our Instructions for Authors page, <http://life-science-alliance.org/authorguide>

Reviewer #1 (Comments to the Authors (Required)):

In this manuscript the authors show that the WNT inhibitor CxxC5 is overall increased in the growth plate as they mature with age. They also show that CxxC5 in chondrocytes is a target of estrogens, potentially linking the end of longitude growth at puberty to induction of CxxC5 expression. Finally they show that disrupting the DVL-Cxxc5 interaction, responsible for inhibition of Wnt signaling, with a small molecule they identified can prolong longitudinal growth. lastly they show that deletion of CxxC5 leads to prolonged longitudinal bone growth. They conclude that the molecules they identified could be used to favor bone growth in patients with growth retardation.

Critique: Overall this is an interesting paper reporting novel data that is translationally relevant. The conclusions are overall supported by the data and the quality of the figures is acceptable. The main weaknesses are however that 1) Nowhere in the manuscript do the authors report the precise localisation of *cxxc5* expression in the growth plate. This is of the utmost importance since the maturation of the growth plate occurs not only with age (as mostly studied here) but also within the growth plate itself through the progression of pre-chondrocytes to proliferation, pre-hypertrophy and hypertrophy. Thus, it is essential to determine the course of *cxxc5* expression during the stepwise progression of chondrocytes in the GP. 2) The authors use Runx2 as a "chondrogenic" marker and report its repression by *cxxc5* as an illustration of "senescence"; this is quite misleading because the authors ignore the most important fact that Runx2 actually needs to be repressed in order to ensure a slow and progressive transition to hypertrophy; they ignore this important fact in their data collection and in their discussion, and they ignore key players such as PTHrP, Indian Hedgehog and FGF18, all key regulators of GP maturation and bone growth within the GP. To present Runx2 simply as a factor that needs to be elevated for GP health is truly misleading. 3) The use of the term senescence (although they do refer to previous papers to support this choice, is also misleading now that this term defines a precise cellular mechanism that is actually not studied here. They may want to use the term "maturation" instead.

Specific comments:

- 1- Fig 1A: although nice to show relative changes with time, it would be useful to have expression levels and localization of the expression within the growth plate and not just in total GP. This tissue is too heterogeneous when it comes to chondrocytes to be treated globally.
- 2- In all these figures one would also want to see expression levels within the different layers of the GP and not only with age.
- 3- Fig 1E: Where are we in the GP? These images are confusing because they fail to indicate which layer we are looking at. They should be retaken to include the whole GP.
- 4- Fig 1F: The authors should include PTHrP, IHH and FGF18 in their analysis of genes in the GP that are directly relevant to GP and longitudinal growth (see Kronenberg HM papers)
- 5- Fig 1D is not acceptable: the E2 section is too clique to be comparable to the control
- 6- Fig 2,3, and 5: unclear where in the GP are some pictures taken. Please specify or show on the low mag where the high mags are taken. In any case these suggest that expression is not homogeneous in the GP.
- 7- Discussion: need to include the work of Kronenberg, Karaplis, Ornitz, etc...on PTHrP, IHH, FGFs and the regulation of Runx2 to prevent hypertrophy in order to prolong the growth...and how does this fit with the current findings. This is why it is essential to analyze precise expression localization within the different layers of the GP.

Reviewer #2 (Comments to the Authors (Required)):

The authors report a novel compound, KY19382, that inhibits both the DVL-PDZ domain and GSK-3 β . If it is true, the study is of biological significance. However, based on the data present in the manuscript, I am not convinced that is the case. Moreover, the authors describe the studies of applying this compound in mice without providing any ADME/PK information. Furthermore, all the necessary control studies are not reported in the manuscript.

Reviewer #3 (Comments to the Authors (Required)):

Short Summary

This manuscript addresses the role of CXXC5, a negative regulator of Wnt/beta-Catenin signaling, on growth plate senescence and associated longitudinal bone growth. Authors used a peptide PTD-DBMP, and a small molecule KY19382, which interferes with GSK-3b activity and CXXC5-DVL interaction. Such interference activates Wnt/beta-catenin signaling that delays growth plate senescence, resulting in longitudinal bone growth in mice; thereby proposing CXXC5 as a potential target to enhance longitudinal growth of children at the risk of height retardation. This is a well described study that needs some more evidences from supporting experiments.

Specific points

1. Power analysis is strongly suggested. Most of the experiments have a low N (n=3). Additional independent experiments will provide power to the study.
2. All the results and corresponding figures state Runx2, Col10a1, MMP13, Alp etc molecules as 'chondrogenic markers' while these molecules are mostly direct osteogenic markers. Including the observations on direct chondrogenic markers such as Sox9, Col2a1, Aggrecan in results and corresponding figures would be more relevant to the scope of this manuscript.
3. Fig 4E showed inactivation of GSK-3beta upon KY19382 treatment. What is its effect on GSK-3alpha? Justify your observations.
4. What is the topological phenotype of elongated bones in CXXC5^{-/-} mice compared to CXXC5^{+/+} controls in the presence and absence of the peptide and KY19382?
5. Real time PCRs are performed using saturating template cDNA concentration (1µl of 2µg/40µl cDNA reaction, was amplified in 10µl iQ SYBR Green Supermix) per real time PCR reaction as described in Materials and methods. Real time PCR data from diluted cDNA template concentration would be interesting; else justify the use of higher template concentration.
6. Relative quantitation of immunoblots is mostly missing. For example, 1D, 4E, 4F, 5E.

Additional issues

1. Scatter plots should be used instead of bar graphs.
2. Statistical analyses: Analysis of variance with post hoc test for multiple comparisons is suggested, wherever applicable.
3. For some experiments, we don't know the N. For example, most of the immunoblots (Fig 1D, 4E, 4F, 5E); radiographs (3A); IHC (3F) etc.

Above mentioned revisions can be completed in a timeframe of 3-4 months.

Reviewer #1

In this manuscript the authors show that the WNT inhibitor CxxC5 is overall increased in the growth plate as they mature with age. They also show that CxxC5 in chondrocytes is a target of estrogens, potentially linking the end of longitude growth at puberty to induction of CxxC5 expression. Finally they show that disrupting the DVL-Cxxc5 interaction, responsible for inhibition of Wnt signaling, with a small molecule they identified can prolong longitudinal growth. Lastly they show that deletion of CxxC5 leads to prolonged longitudinal bone growth. They conclude that the molecules they identified could be used to favor bone growth in patients with growth retardation.

Critique:

- 1) **Nowhere in the manuscript do the authors report the precise localization of cxxc5 expression in the growth place.**

This is of the utmost importance since the maturation of the growth plate occurs not only with age (as mostly studied here) but also within the growth plate itself through the progression of pre-chondrocytes to proliferation, pre-hypertrophy and hypertrophy. Thus, it is essential to determine the course of cxxc5 expression during the stepwise progression of chondrocytes in the GP. Response:

Response: To address this issue, we now provided new data for CXXC5 expression in different zones of the growth plate in Fig 1E, 2E, and 2F of the revised manuscript. We found that CXXC5 was expressed in all resting, proliferative, and hypertrophic zones at late puberty. In addition, the tibial growth plates treated with E2 both *ex vivo* and *in vivo* showed enhanced CXXC5 expression in chondrocytes of proliferative and hypertrophic

zones with decreased height of each zone. Furthermore, proliferation and differentiation of chondrocytes were increased in the growth plates of *Cxxc5*^{-/-} mice compared with those of wild type mice. We describe and discuss these observations in the Discussion section of the revised manuscript (page 13, lines 9–12).

- 2) **The authors use Runx2 as a "chondrogenic" marker and report its repression by cxxc5 as an illustration of "senescence"; this is quite misleading because the authors ignore the most important fact that Runx2 actually needs to be repressed in order to ensure a slow and progressive transition to hypertrophy; they ignore this important fact in their data collection and in their discussion, and they ignore key players such as PTHrP, Indian Hedgehog and FGF18, all key regulators of GP maturation and bone growth within the GP. To present Runx2 simply as a factor that needs to be elevated for GP health is truly misleading.**

Response: We appreciate the reviewer's critiques addressing the relationship between hypertrophic differentiation and cessation of bone growth. We understand the reviewer's comments to be based on the hypothesis that growth plate closure causes growth cessation. Growth plate closure occurs through the replacement of cartilage by bone tissue and requires chondrocyte hypertrophy, apoptosis, and ossification without proliferation. However, by careful observation, it was shown that longitudinal bone growth ceases with the suppression of both proliferation and hypertrophic differentiation of chondrocytes at late puberty prior to growth plate closure, which implies growth plate closure is not the cause but the result of growth cessation (Moss & Noback, 1958; Nilsson & Baron, 2004; Weise et al, 2001). This phenomenon, called "growth plate senescence", is consistent with our current observation of gradual reduction of chondrocyte

proliferation and differentiation with pubertal progression. However, to avoid confusion and to effectively present our data showing the time-course changes of Wnt/ β -catenin signaling and CXXC5 expression in Fig 1, we have removed the RUNX2 data in Fig 1C and 1E of the revised manuscript. Furthermore, as the reviewer suggested, we have now measured the expression pattern of FGF18, IHH, and PTHrP following WNT3A treatment with or without CXXC5 overexpression (Fig. S3). We describe and discuss these data in the Results and Discussion sections of the revised manuscript, respectively (page 6, lines 3–7 and page 13, lines 12–15).

- 3) The use of the term senescence (although they do refer to previous papers to support this choice, is also misleading now that this term defines a precise cellular mechanism that is actually not studied here. They may want to use the term "maturation" instead.

Response: We thank the reviewer for the suggestion; however, the term “growth plate senescence” has been used in many articles to address the structural senescence of the growth plate that occurs with increasing age. In this context, “growth plate senescence” indicates gradual reduction of the height of resting, proliferative, and hypertrophic zones along with decreased numbers of proliferative and hypertrophic chondrocytes (Lui et al, 2011; Nilsson & Baron, 2004; Weise et al, 2001). The term “maturation” merely implies the promotion of chondrocyte hypertrophy and subsequent ossification without consideration of aging. Furthermore, our results demonstrate suppression of chondrocyte proliferation and hypertrophic differentiation in the growth plate with pubertal progression. Therefore, we consider the term “growth plate senescence” to be more suitable for our study.

Specific comments:

- 1. Fig 1A: although nice to show relative changes with time, it would be useful to have expression levels and localization of the expression within the growth plate and not just in total GP. This tissue is too heterogeneous when it comes to chondrocytes to be treated globally.**

Response: The data presented in Fig 1A represent the gene expression profiles of the Wnt/ β -catenin pathway-related genes in the proliferative zones of the rat growth plate from 3- to 12-weeks of age. Unfortunately, time-course gene expression in the resting and hypertrophic zones are not available in the database. We have clarified the description of the specific zone of the growth plate in the figure legend of the revised manuscript (Fig 1).

- 2. In all these figures one would also want to see expression levels within the different layers of the GP and not only with age.**

Response: We have replaced the original data with low magnification image data in the figures to show the whole growth plate area in the revised manuscript (Fig 1E, 2E, 2F, 3I, and 5A).

- 3. Fig 1E: Where are we in the GP? These images are confusing because they fail to indicate which layer we are looking at. They should be retaken to include the whole GP.**

Response: We have replaced the original data with low magnification images to show the

whole growth plate. To clearly display the localization of CXXC5, we have also included magnified images of each zone (Fig 1E). During the progression of growth plate senescence, expression levels of CXXC5 and β -catenin were increased and decreased, respectively, in all growth plate zones with aging.

- 4. Fig 1F: The authors should include PTHrP, IHH and FGF18 in their analysis of genes in the GP that are directly relevant to GP and longitudinal growth (see Kronenberg HM papers)**

Response: For better presentation of our results, Fig 1F of the original manuscript has been moved to Fig S3 of the revised manuscript. As the reviewer requested, we have now analyzed the transcriptional levels of FGF18, IHH, and PTHrP, and present these results in the revised manuscript (Fig S3). FGF18 is a direct target gene of the Wnt/ β -catenin pathway (Reinhold & Naski, 2007). Although IHH and PTHrP are not Wnt/ β -catenin pathway target genes, RUNX2, a direct target of the Wnt/ β -catenin pathway, promotes IHH expression, which in turn stimulates PTHrP production (Pratap et al, 2008; Tian et al, 2014). Consistent with these relationships, mRNA expression levels of *Fgf18*, *Ihh*, and *Pthlh* were induced in a Wnt/ β -catenin signaling-dependent manner as shown in Fig S3 of the revised manuscript.

- 5. Fig 1D is not acceptable: the E2 section is too clique to be comparable to the control.**

Response: We have replaced the problematic original data for the Fig 1D with the new data in the revised manuscript.

6. **Fig 2, 3, and 5: unclear where in the GP are some pictures taken. Please specify or show on the low mag where the high mags are taken. In any case these suggest that expression is not homogeneous in the GP.**

Response: We have replaced the original data with low magnification images in Fig 2, 3, and 5 in the revised manuscript to show the entire growth plate. The data now clearly indicate that CXXC5, which is expressed in all growth plate zones, suppresses both proliferation and hypertrophic differentiation of chondrocytes. Furthermore, abrogation of CXXC5 function promotes longitudinal bone growth by delaying growth plate senescence.

7. **Discussion: need to include the work of Kronenberg, Karaplis, Ornitz, etc...on PTHrP, IHH, FGFs and the regulation of Runx2 to prevent hypertrophy in order to prolong the growth...and how does this fit with the current findings. This is why it is essential to analyze precise expression localization within the different layers of the GP.** **Response:** As suggested by the reviewer, we now discuss the effects of Wnt/ β -catenin signaling and CXXC5 on the expression of PTHrP, IHH, and FGF18 in the Discussion section of the revised manuscript (page 13, lines 9–15). We appreciate the reviewer's professional comments regarding hypertrophy and bone growth; however, our findings show that retardation of longitudinal bone growth occurred with the decline of both chondrocyte proliferation and differentiation. Our results thus indicate that growth plate chondrocytes may have already undergone suppression of overall chondrogenic processes prior to growth plate closure.

Reviewer #2

The authors report a novel compound, KY19382, that inhibits both the DVL-PDZ domain and GSK-3beta. If it is true, the study is of biological significance. However, based on the data present in the manuscript, I am not convinced that is the case. Moreover, the authors describe the studies of applying this compound in mice without providing any ADME/PK information. Furthermore, all the necessary control studies are not reported in the manuscript.

Response: We appreciate the reviewer’s interest related to the drug development of KY19382; however, we believe that the primary outcome of this study is the identification of CXXC5 as a new factor in growth plate senescence and as a potential therapeutic target for longitudinal bone growth. Nevertheless, we have now included the basic PK information for KY19382 (Table R1). We have also included the initial *in vitro* toxicity tests, which reveal that KY19382 does not induce cytotoxicity in variety of cell types (Table R2).

PK Parameters	IV, 1 mg/kg		IP, 5 mg/kg	
	mean	SD	mean	SD
t_{max} (hr)	N/A	-	1.00	0.00
C_{max} (ng/mL)	N/A	-	463.37	29.41
AUC_{last} (ng*hr/mL)	7832.81	651.28	6555.79	572.85
CL (L/hr/kg)	0.12	0.01	0.47	0.03
V_{ss} (L/kg)	0.33	0.07	N/A	
t_{1/2} (hr)	3.33	1.34	16.20	3.86
F (%)	N/A	-	16.74	-

Table R1. Pharmacokinetic profiles for KY19382.

Pharmacokinetic parameters were based on the mean plasma concentration-time profiles of SD male rat (n=3). Pharmacokinetic parameters were obtained by non-compartmental analysis of the plasma concentration-time profiles using Kinetica™ 4.4.1 (Thermo Fisher Scientific, Inc., Woburn, MA, USA). AUC_{last} was calculated from 0 to 24 hour. IV,

intravenous; IP, intraperitoneal; T_{max} , Time to maximum plasma concentration; C_{max} , Maximum plasma concentration after intraperitoneal injection; AUC, Area under the curve; CL, clearance; V_{ss} , Volume of distribution at steady state; $T_{1/2}$, Elimination half-life; F, bioavailability.

Compound	IC ₅₀ (μM)				
	VERO	HFL-1	L929	NIH 3T3	CHO-K1
KY19382	>100	>100	>100	>100	>100

Table R2. Cytotoxicity test for KY19382

Various cells were treated with 0.01, 0.1, 1, 10, 100 μM KY19382. After 24 hours, cell viability was measured using WST-1. The IC₅₀ value was determined from the dose-response curve.

Reviewer #3

Short Summary:

This manuscript addresses the role of CXXC5, a negative regulator of Wnt/beta-Catenin signaling, on growth plate senescence and associated longitudinal bone growth. Authors used a peptide PTD-DBMP, and a small molecule KY19382, which interferes with GSK-3b activity and CXXC5-DVL interaction. Such interference activates Wnt/beta-catenin signaling that delays growth plate senescence, resulting in longitudinal bone growth in mice; thereby proposing CXXC5 as a potential target to enhance longitudinal growth of children at the risk of height retardation. This is a well

described study that needs some more evidences from supporting experiments.

Specific points:

- 1. Power analysis is strongly suggested. Most of the experiments have a low N (n=3).**

Additional independent experiments will provide power to the study.

Response: As the reviewer's comments, we additionally performed experiments for Fig 1C, 1E, and 3I. We have provided the results with detailed information for N number in the figure legends of the revised manuscript.

- 2. All the results and corresponding figures state Runx2, Col10a1, MMP13, Alp etc molecules as 'chondrogenic markers' while these molecules are mostly direct osteogenic markers. Including the observations on direct chondrogenic markers such as Sox9, Col2a1, Aggrecan in results and corresponding figures would be more relevant to the scope of this manuscript.**

Response: RUNX2, COL10A1, MMP13, and ALP are well-established markers for hypertrophic differentiation of late chondrocytes (Goldring et al, 2006). Our purpose was to elucidate the effects of CXXC5 on proliferation and hypertrophy of chondrocytes in the growth plate at the pubertal stage, not at embryonic or neonatal stages. Therefore, we focused on the markers RUNX2, COL10A1, MMP13 and ALP, instead of SOX9 and COL2A1, which are important for formation of the growth plate as determinants of chondrogenic lineage at early developmental stages (Bi et al, 1999; Kozhemyakina et al, 2015). However, as the reviewer suggested, we analyzed the transcriptional expression of SOX9 and COL2A1, and the results of this analysis are now presented in Fig S3 of the

revised manuscript. Additionally, we have confirmed the expression level of COL2A1 in other figures, including Fig 1D, 3G, 5A, 5E, 5F, S7B, and S7C. Our results indicate that expression of diverse chondrogenic markers are regulated by CXXC5 or KY19382.

3. Fig 4E showed inactivation of GSK-3beta upon KY19382 treatment. What is its effect on GSK-3alpha? Justify your observations.

Response: To address this query, we detected the inactive forms of GSK3 α/β using a phospho-GSK3 α/β (Ser21/9) antibody. The results demonstrated that KY19382 effectively inactivated both GSK3 α and GSK3 β . These new results have replaced the original data in the revised manuscript (Fig 4E).

4. What is the topological phenotype of elongated bones in CXXC5^{-/-} mice compared to CXXC5^{+/+} controls in the presence and absence of the peptide and KY19382?

Response: The peptide is predicted to be ineffective in *Cxxc5^{-/-}* mice, because it is a CXXC5 inhibitor. On the other hand, KY19382 is expected to partially enhance longitudinal bone growth in *Cxxc5^{-/-}* mice, because KY19382 can activate Wnt/ β -catenin signaling by inhibition of GSK3 β independently of CXXC5. Although we did not perform studies using the peptide and KY19382 in *Cxxc5^{-/-}* mice, we did identify the effects of each compound on growth plate senescence and longitudinal bone growth in wild type mice. In addition, the functional mechanism of KY19382 was demonstrated using *in vitro* analyses of both CXXC-DVL binding and GSK3 β kinase activity (Fig 4B and 4C) as well as by western blot and immunoprecipitation analyses (Fig 4E and 4F). Finally, the mechanism and effectiveness of the peptide has been verified in our previous

studies, which show that the peptide interferes with CXXC5 function by blocking its interaction with DVL in *in vitro* and *in vivo* models (Kim et al, 2015; Lee et al, 2015; Lee et al, 2017).

- 5. Real time PCRs are performed using saturating template cDNA concentration (1 μ l of 2 μ g/40 μ l cDNA reaction, was amplified in 10 μ l iQ SYBR Green Supermix) per real time PCR reaction as described in Materials and methods. Real time PCR data from diluted cDNA template concentration would be interesting; else justify the use of higher template concentration.**

Response: We thank the reviewer for pointing out this misleading description in our methodology. We have modified the description in the revised manuscript to state that 1 μ l of 5–100-fold diluted cDNA was used for real-time PCR analysis (page 21, line 12).

- 6. Relative quantitation of immunoblots is mostly missing. For example, 1D, 4E, 4F, 5E.**

Response: We quantified the immunoblot bands and provide the numbers in Fig 1D, 4E, 4F, and 5E of the revised manuscript.

Additional issues:

- 1. Scatter plots should be used instead of bar graphs.**

Response: As the reviewer's suggestion, we have replaced the original data with scatter plot form in Fig 1B and 5J of the revised manuscript.

- 2. Statistical analyses: Analysis of variance with post hoc test for multiple comparisons is suggested, wherever applicable.**

Response: We newly performed ANOVA and followed Tukey's or Bonferroni's post-hoc test for more than two groups in Fig 1B, 1C, 1E, 3B, 3D, S3, S7A, and S8 of the revised manuscript.

- 3. For some experiments, we don't know the N. For example, most of the immunoblots (Fig 1D, 4E, 4F, 5E); radiographs (3A); IHC (3F) etc.**

Response: As the reviewer guided, we provided the N numbers for experiments, and described in the Statistical analyses part of Materials and Methods section in the revised manuscript. The results represent at least three independent experiments.

Reference

Bi W, Deng JM, Zhang Z, Behringer RR, de Crombrughe B (1999) Sox9 is required for cartilage formation. *Nat Genet* 22: 85-89

Goldring MB, Tsuchimochi K, Ijiri K (2006) The control of chondrogenesis. *J Cell Biochem* 97: 33-44

Kim HY, Yoon JY, Yun JH, Cho KW, Lee SH, Rhee YM, Jung HS, Lim HJ, Lee H, Choi J et al (2015) CXXC5 is a negative-feedback regulator of the Wnt/beta-catenin pathway involved in osteoblast differentiation. *Cell Death Differ* 22: 912-920

Kozhemyakina E, Lassar AB, Zelzer E (2015) A pathway to bone: signaling molecules and transcription factors involved in chondrocyte development and maturation.

Development 142: 817-831

Lee SH, Kim MY, Kim HY, Lee YM, Kim H, Nam KA, Roh MR, Min do S, Chung KY, Choi KY (2015) The Dishevelled-binding protein CXXC5 negatively regulates cutaneous wound healing. *J Exp Med* 212: 1061-1080

Lee SH, Seo SH, Lee DH, Pi LQ, Lee WS, Choi KY (2017) Targeting of CXXC5 by a Competing Peptide Stimulates Hair Regrowth and Wound-Induced Hair Neogenesis. *J Invest Dermatol* 137: 2260-2269

Lui JC, Nilsson O, Baron J (2011) Growth plate senescence and catch-up growth. *Endocr Dev* 21: 23-29

Moss ML, Noback CR (1958) A longitudinal study of digital epiphyseal fusion in adolescence. *Anat Rec* 131: 19-32

Nilsson O, Baron J (2004) Fundamental limits on longitudinal bone growth: growth plate senescence and epiphyseal fusion. *Trends Endocrinol Metab* 15: 370-374

Pratap J, Wixted JJ, Gaur T, Zaidi SK, Dobson J, Gokul KD, Hussain S, van Wijnen AJ, Stein JL, Stein GS et al (2008) Runx2 transcriptional activation of Indian Hedgehog and a downstream bone metastatic pathway in breast cancer cells. *Cancer Res* 68: 7795-7802

Reinhold MI, Naski MC (2007) Direct interactions of Runx2 and canonical Wnt signaling induce FGF18. *J Biol Chem* 282: 3653-3663

Tian F, Wu M, Deng L, Zhu G, Ma J, Gao B, Wang L, Li YP, Chen W (2014) Core binding factor beta (Cbfbeta) controls the balance of chondrocyte proliferation and differentiation by upregulating Indian hedgehog (Ihh) expression and inhibiting

parathyroid hormone-related protein receptor (PPR) expression in postnatal cartilage and bone formation. *J Bone Miner Res* 29: 1564-1574

Weise M, De-Levi S, Barnes KM, Gafni RI, Abad V, Baron J (2001) Effects of estrogen on growth plate senescence and epiphyseal fusion. *Proc Natl Acad Sci U S A* 98: 6871-6876

March 25, 2019

RE: Life Science Alliance Manuscript #LSA-2018-00254-TR

Dear Dr. Choi,

Thank you for submitting your revised manuscript entitled "CXXC5 mediates growth plate senescence and is a target for enhancement of longitudinal bone growth". As you will see, the reviewers appreciate the introduced changes and we would thus be happy to publish your paper in Life Science Alliance pending final revisions necessary to address reviewer #2's comment regarding the ADME/PK data. Please include those in the final version of the manuscript as well as the further discussion as suggested by the reviewer. Please also link your profile in our submission system to your ORCID iD, you should have received an email with instructions on how to do so.

A. FINAL FILES:

B. MANUSCRIPT ORGANIZATION AND FORMATTING:

We encourage our authors to provide original source data, particularly uncropped/-processed

electrophoretic blots and spreadsheets for the main figures of the manuscript. If you would like to add source data, we would welcome one PDF/Excel-file per figure for this information. These files will be linked online as supplementary "Source Data" files.

Sincerely,

Reviewer #1 (Comments to the Authors (Required)):

Overall, the authors have addressed all my concerns and provided acceptable explanations to some of my comments. I think the manuscript has been much improved by the revisions and should now be publishable in LSA.

Reviewer #2 (Comments to the Authors (Required)):

Because the study used a novel compound in the mouse study, therefore, it is important to know the ADME/PK information of the compound, it has nothing to do with drug development.

Nevertheless, I am glad that the authors carried out the study. However, the relevant ADME/PK data should be incorporated into the manuscript.

Also, although the $t_{1/2}$ (16 hrs) seems to fit very well with the experimental design (daily IP injection), the C_{max} (about 0.03 μM) looks a little bit low compared to the in vitro studies (e.g., 0.1 μM in Fig. 4D and 4G). The authors should discuss this issue as well.

Reviewer #3 (Comments to the Authors (Required)):

Manuscript describes the role of CXXC5, a negative regulator of Wnt/betaCatenin signaling, on growth plate senescence and associated longitudinal bone growth. Authors used a peptide PTD-DBMP, and a small molecule KY19382, which interferes with GSK-3 β activity and CXXC5-DVL interaction. This interference activates Wnt/beta-catenin signaling that delays growth plate senescence, resulting in longitudinal bone growth in mice; thereby proposing CXXC5 as a potential target to enhance longitudinal growth of children at the risk of height retardation. Revised manuscript addressed the concerns raised during first round of revisions and included supporting evidences in results, figures and wherever necessary. Therefore, this article can be considered for acceptance in Life Science Alliance.

Thank you very much for your comments for our manuscript. We were pleased to find that

Reviewer #1 and Reviewer #3 were satisfied with our revised manuscript. In response to the

suggestion made by Reviewer #2, we have included the basic PK information for KY19382 in the 2nd revised manuscript (Table S3). Although we understand the reviewer's concern that the C_{max} of *in vivo* pharmacokinetics for KY19382 looks a little bit low compared to the *in vitro* studies, we confirmed that 0.1 mg/kg KY19382 was most effective for the inhibition of growth plate senescence in preliminary *in vivo* study that was tested by administration of 0.005, 0.01, 0.05, 0.1, and 0.5 mg/kg KY19382 (Figure R2-1; below). In addition, we identified that 0.01 μ M KY19382 was sufficient to promote the activation of Wnt/ β -catenin pathway *in vitro* level (Fig 4E and Fig S8). The optimal concentration of KY19382 might be different for *in vitro* and *in vivo* effectiveness. We described these data in the Results section of the 2nd revised manuscript (page 12, lines 3–5). Moreover, we re-formatted the manuscript as the editor recommended to fit the guidelines provided by *Life Science Alliance*.

April 1, 2019

RE: Life Science Alliance Manuscript #LSA-2018-00254-TRR

Dear Dr. Choi,

Thank you for submitting your Research Article entitled "CXXC5 mediates growth plate senescence and is a target for enhancement of longitudinal bone growth". It is a pleasure to let you know that your manuscript is now accepted for publication in Life Science Alliance. Congratulations on this interesting work.

DISTRIBUTION OF MATERIALS:

Again, congratulations on a very nice paper. I hope you found the review process to be constructive and are pleased with how the manuscript was handled editorially. We look forward to future exciting submissions from your lab.

Sincerely,

Andrea Leibfried, PhD
Executive Editor
Life Science Alliance